# A Novel Sample-to-Answer Visual Nucleic Acid Detection System for Adenovirus Detection

Kui Sun,[a,b] Xiaodong Yang,[c] Yanan Wang,[a,d] Qun Guan,[e] Wenliang Fu,[a] Chao Zhang,[a] Qin Liu,[a] Wenzheng An,[a] Yongqi Zhao,[a] Weiwei Xing,[a] Donggang Xu[a]

[a]Beijing Institute of Basic Medical Sciences, Beijing, China
[b]Energy Laboratory of 970 Hospital of the PLA Joint Logistic Support Force, Beijing, China
[c]Department of General Surgery, the First Medical Center of Chinese PLA General Hospital, Beijing, China
[d]Academy of Medical Laboratory, Hebei North University, Zhangjiakou, China
[e]The Fifth Medical Center of Chinese PLA General Hospital, Beijing, China

Kui Sun, Xiaodong Yang and Yanan Wang contributed equally to this work. Author order was determined by seniority.

**ABSTRACT** Human adenoviruses (HAdVs) are common viruses that can cause local outbreaks in schools, communities and military camps, posing a huge threat to public health. An ideal POCT device for adenovirus detection in resource-limited settings is critical to control the spread of the virus. In this study, we developed an integrated and electricity-independent sample-to-answer system that can complete nucleic acid extraction, amplification, and detection at room temperature. This system is suitable for field and on-site detection because of its rapidity, sensitivity, lack of contamination, and lack of requirements of high-precision instruments and skilled technicians. It consists of two separate modules, ALP FINA (alkaline lysis with the paper-based filtration isolation of nucleic acid) and SV RPA (sealed and visual recombinase polymerase amplification). The extraction efficiency of ALP FINA can reach 48 to 84%, which is close to that of the conventional centrifuge column. The detection sensitivity of SV RPA is close to 10 copies/$\mu$L of AdvB and AdvE without aerosol contamination after repeated operations. When SV RPA was applied to the detection of nasopharyngeal swab samples of 19 patients who were infected with AdvB or AdvE as well as 10 healthy volunteers, its sensitivity and specificity reached 100%, respectively.

**IMPORTANCE** HAdV infections are readily transmittable and, in some instances, highly contagious. Early and rapid diagnosis is essential for disease control. In this work, we developed a portable, disposable, and modularized sample-to-answer detection system for AdvB and AdvE, which rendered the entire test to be completely independent of electricity and other laboratory infrastructure. Thus, this detection system can be applied in resource-limited settings, and it has the potential to be further developed as an early diagnosis method in the field.

**KEYWORDS** RPA, sample to answer, human adenoviruses, aerosol contamination, NAAT

Human adenovirus (HAdV) is a common infectious pathogen that can cause large outbreaks in closed environments, especially in military camps and schools (1–4). The early diagnosis of adenovirus is critical for controlling the spread of an epidemic. Currently, the detection and identification of adenovirus mainly adopt qPCR (5), which has high sensitivity and specificity. However, this method is not suitable for on-site applications due to its dependence on sophisticated instruments and trained personnel. Other detection methods, such as viral culture, are time-consuming and laborious. Serologic tests show the moderate sensitivity and specificity due to the time lags of antibody appearance and immune cross-reaction. Therefore, a point-of-care based nucleic acid amplification test (NAAT) for human

Address correspondence to Yongqi Zhao, yqzhaoprc@sina.com, Weiwei Xing, huozinangua@163.com, or Donggang Xu, xudg@bmi.ac.cn.

The authors declare no conflict of interest.

adenovirus is essential for the early prevention and control of adenovirus infection. The key to realize the point-of-care testing (POCT) of adenovirus nucleic acid detection relies on rapid nucleic acid extraction, convenient nucleic acid amplification and result determination, as well as the elimination of aerosol pollution. In this study, we developed an integrated and electricity-independent sample-to-answer system that can complete nucleic acid extraction, amplification, and detection at room temperature.

As sample preparation is the first step in nucleic acid-based testing, an extraction method that is simple, rapid, portable, and independent of electricity is essential for field applications of NAAT. The use of paper-based nucleic acid extraction techniques has recently emerged for POCT, involving the use of a specifically designed membrane for filtration by selectively retaining the DNA from lysed samples (6). However, it currently still requires proteases for lysing the samples, which requires a heating step (7). Therefore, in order to remove the dependence on electric equipment, we developed a module that combined alkaline lysis with the paper-based filtration isolation of nucleic acid (ALP FINA). The sample can be sequentially treated with NaOH, absolute ethanol, and nuclease-free water for lysis, and the nucleic acid can be extracted using a filter containing an extraction membrane. The entire extraction process requires neither heating nor centrifugation, and it can be performed within 15 min without relying on electrical facilities.

In addition to sample preparation, the rapid amplification and visual detection of samples is critical to the realization of NAAT-based POCT. In this respect, isothermal amplification technologies can achieve the rapid amplification of pathogens at room temperature (8), among which recombinase polymerase amplification (RPA) technology has outstanding advantages (9). It has been demonstrated to be ideal for resource-limited applications, and it can amplify trace nucleic acids to detectable levels within 15 min over a wide temperature range (10). For visual detection, nucleic acid dyes or lateral flow dipstick (LFD) methods are commonly used to detect amplification products (11–13). Calcein can be added into the reaction procedure to realize the visual judgment of the results (14, 15). However, its addition would reduce the sensitivity of the reaction. In recent years, some LFD-based nucleic acid isothermal amplification techniques have begun to emerge (16). LFD can bind to the amplification products and presents a visible signal, in consequence, which is a simple and convenient operation and exhibits many of the desired characteristics of an ideal visualization method. In particular, the application of LFD does not show any inhibitory effects on the reaction and has been combined with various isothermal amplification technologies (16–20). Thus, the combination of RPA and LFD (RPA-LFD) constitutes a sensitive, specific, rapid, and visual detection system with great potential applicability in POCT (19). The main drawback of traditional RPA-LFD methods is their open operation mode, which increases the possibility of their contamination by aerosolized nucleic acids, which could lead to false positives in subsequent detection. To overcome this problem, we developed a sealed and visual RPA testing device (SV RPA) that integrates the entire RPA-LFD assay into a single device. In particular, SV RPA can conduct the RPA reaction, the delivery of the reaction medium, and the LFD detection within the same sealed space, without the need for opening the lid, which could effectively prevent the possibility of contamination.

To meet all of the requirements for a fully integrated point-of-care NAAT system that is suitable for resource-limited settings, we combined the two independent working modules of ALP FINA and SV RPA into a sample-to-answer system. The ALP FINA module utilizes the alkaline lysis method, combined with a Fusion 5 membrane, to achieve nucleic acid purification, and this does not require heat or centrifugation. The SV RPA module utilizes the visualization advantages of the RPA-LFD assay, which can be efficiently carried out at room temperature, making the entire process electricity-independent. Thus, the system would be suitable for field applications and rapid diagnosis in community clinics.

Adenovirus (Adv) infection is a frequent cause of respiratory diseases in China (21). There are seven types of Advs (types A to G) that lead to a variety of clinical symptoms, among which Types B and E often result in acute febrile respiratory diseases (22). They can cause epidemic outbreaks and are associated with severe symptoms, leading to death in some cases (23). To evaluate the validity of the integrated device, we applied

**TABLE 1** Comparison of the DNA extracted from various concentrations of reference strains via the ALP FINA module and traditional centrifugal column extraction via qPCR

| Input DNA (copies/$\mu$L) | Recovered DNA (copies/$\mu$L) | |
| --- | --- | --- |
| | Centrifugal column | ALP fINA |
| $5 \times 10^9$ | $4.3 \times 10^9$ | $3.9 \times 10^9$ |
| $5 \times 10^8$ | $3.8 \times 10^8$ | $2.9 \times 10^8$ |
| $5 \times 10^7$ | $4.4 \times 10^7$ | $3.3 \times 10^7$ |
| $5 \times 10^6$ | $4.6 \times 10^6$ | $4.2 \times 10^6$ |
| $5 \times 10^5$ | $3.2 \times 10^5$ | $2.5 \times 10^5$ |
| $5 \times 10^4$ | $3.0 \times 10^4$ | $2.4 \times 10^4$ |

this new NAAT system for the detection of clinical samples of AdvB and AdvE, and we evaluated its sensitivity and specificity to determine the effectiveness of the proposed system.

## RESULTS

**Performance of the ALP FINA module.** An analysis of the two DNA extract methods, including the ALP FINA module and traditional centrifugal column extraction via qPCR showed that the DNA recovery of ALP FINA was within the range of 48% to 84%, whereas that of the traditional method was 60% to 92% (Table 1). The correlations of the input DNA amount and the recovered DNA were high for both the ALP FINA module and the centrifugation method at $r^2$ = 0.931 and 0.992, respectively. Despite the correlation of the traditional method being slightly stronger than that of the ALP FINA module, the extraction efficiencies of the two methods were not significantly different. (Fig. 1).

**Performance of the SV RPA module. (i) Sensitivity and specificity.** The serial dilutions of AdvB and AdvE DNA from $10^6$ copies/$\mu$L to 1 copy/$\mu$L were used for the threshold detection of the SV RPA and qPCR. The results demonstrated that AdvB and AdvE could be simultaneously detected via SV RPA, using the universal primer pairs with the same sensitivity of 10 copies/$\mu$L, and these results was similar to those of the qPCR method (Fig. 2). The probit regression model was used to calculate the detection limits of the SV RPA module for both virus subtypes (AdvB and AdvE). The limits of detection of AdvB and AdvE were 14.261 copies/$\mu$L (95% CI, 13.013 to 16.279) and 13.604 copies/$\mu$L (95% CI, 12.382 to 15.550), respectively. A specificity analysis showed that the SV RPA for AdvB and AdvE had good abilities to discriminate their target DNA from DNA of other species of viruses (AdvC, AdvD, H1N1, and norovirus). Except for the AdvB and AdvE samples, the red-purple-colored line was only observed as the control line on the LFD strips (Fig. 3A).

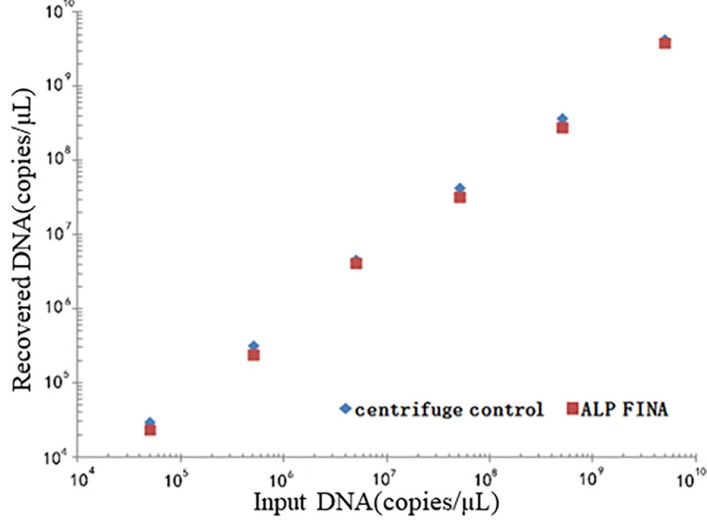

**FIG 1** The correlations of the input DNA amount and recovered DNA.

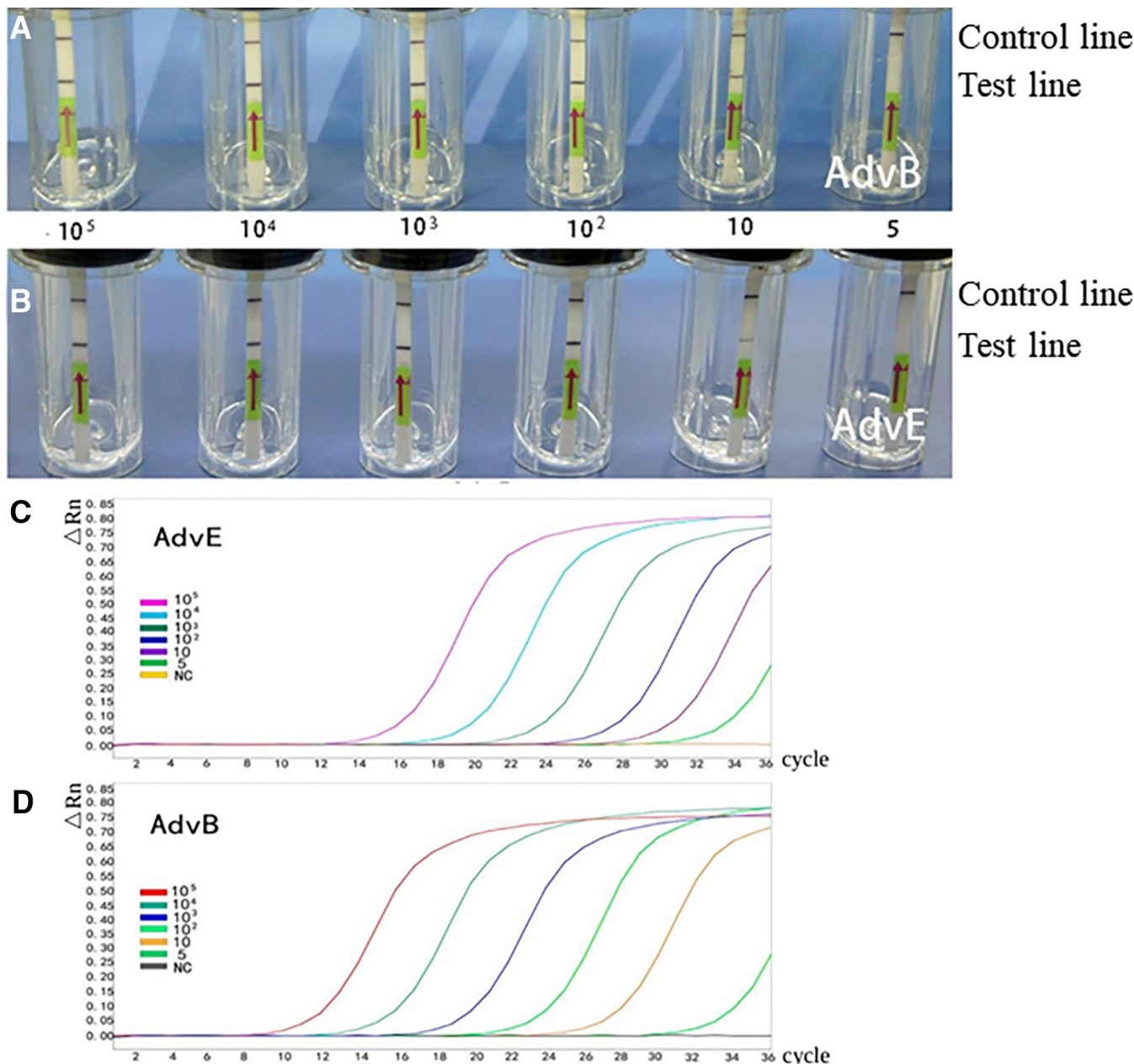

**FIG 2** Detection sensitivity of SV RPA and qPCR. (A) Detection limit of AdvB using SV RPA. (B) Detection limit of AdvE using SV RPA. (C) Detection limit of AdvB using qPCR. (D) Detection limit of AdvB using qPCR.

**(ii) Temperature and time optimization.** The results showed that the detection could be effectively operated at a range of temperatures from 25℃ to 45℃ (Fig. 3B). The intensity of the band was darkest at 35 to 45℃, indicating the optimal temperature. With respect to the reaction time, a positive band, although relatively light, began to appear at 5 min, and it then became significantly darker after 10 min (Fig. 3C). To balance the maximum sensitivity and detection speed for use at the point-of-care, we recommend an amplification time of 10 to 15 min for future experiments.

**(iii) Sealing.** There were no false-positive results in multiple repetitive experiments with SV RPA that were conducted in a separate room. In contrast, the traditional open mode of the RPA-LFD assay yielded false-positive results in the negative control in the third and subsequent experiments. This result demonstrated that the SV RPA device has reliable sealing that could effectively prevent aerosol contamination (Table 2).

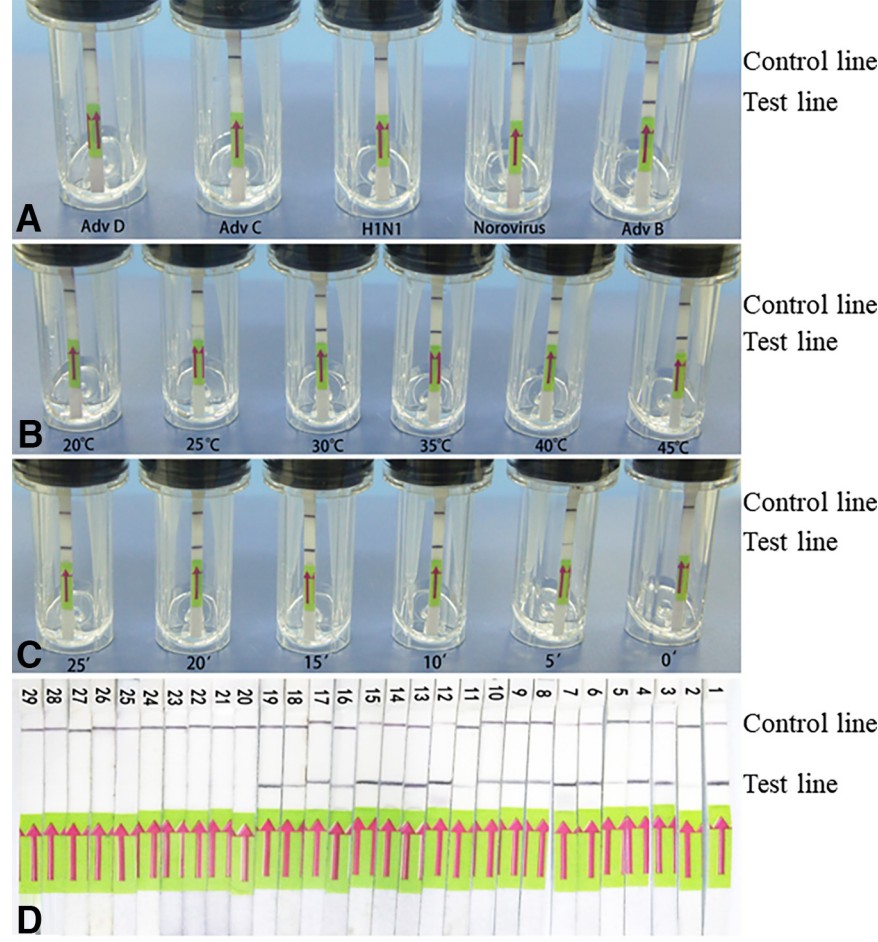

**FIG 3** Specificity and condition optimization of SV RPA. (A) Detection specificity of AdvB using SV RPA. (B) Temperature optimization of SV RPA. (C) Reaction time optimization of SV RPA. (D) The LFD data for tests on positive and negative samples.

**Sample-to-answer application in clinical samples.** As shown in Fig. 3D, all 19 nasopharyngeal swab samples from the patients who were infected with AdvB or AdvE yielded a positive result with the SV RPA device, whereas the 10 samples from healthy volunteers showed negative results.

## DISCUSSION

At present, the POCT for detecting pathogens has been widely used in clinical practice, such as in the antigen detection of SARS-CoV-2. Although antigen detection showed high specificity, its sensitivity could not reach similar a level to that of NAAT. NAAT is crucial for the prevention and control of infectious diseases due to its high sensitivity. Currently, routine NAAT relies mainly on professional personnel in a laboratory setting using specialized instruments, which limits disease screening in resource-poor areas. Therefore, the POCT based on NAAT is the trend for future pathogen detection on-site, among which the concept of sample-to-answer is highly respected. An ideal "sample-to-answer" should have the following "advantages" high sensitivity, good specificity, a convenient and efficient operation process

**TABLE 2** Comparing the seal of SV RPA and traditional RPA-LFD

| Operation method Control | First | | Second | | Third | | Fourth | |
|---|---|---|---|---|---|---|---|---|
| | Positive | Negative | Positive | Negative | Positive | Negative | Positive | Negative |
| SV RPA | + | − | + | − | + | − | + | − |
| Traditional RPA-LFD | + | − | + | − | + | + | + | + |

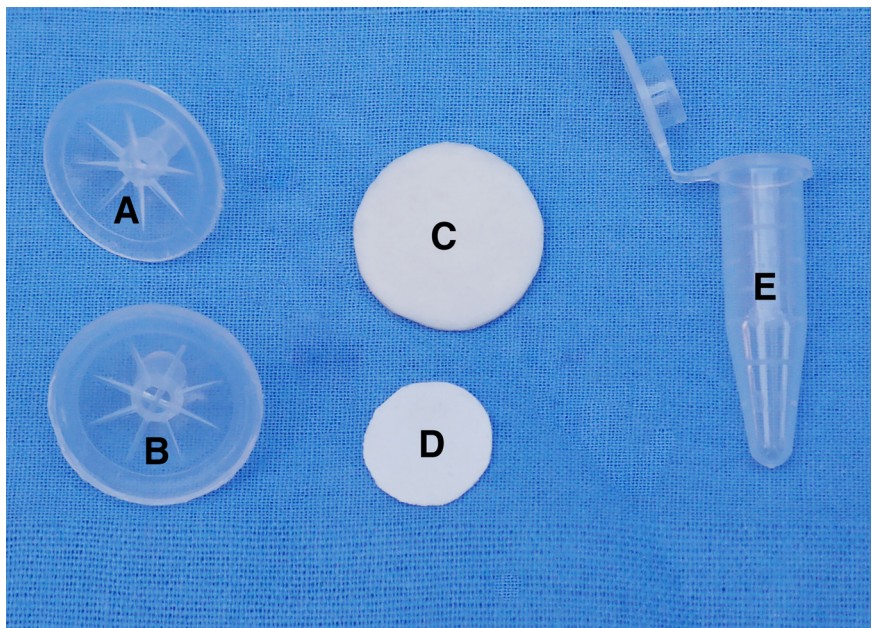

**FIG 4** Exploded view of the ALP FINA prototype. (A) Bottom cover. (B) Top cover. (C) Cellulose filter paper. (D) Fusion 5 filter paper. (E) 1.5 mL tube.

(including nucleic acid extraction, amplification, and detection), an independence from expensive equipment, trained personnel, and external power, and no contamination by aerosolized nucleic acid. However, the current methods of realizing sample-to-answer mostly apply to microfluidic strategies. Wang et al. proposed an automated microfluidic system that depended on PCR. The system integrates the sample preparation area, amplification area, and product analysis area into one sample channel. The sensitivity of the system to detect HPV is 200 copies/test (24). Tian et al. developed an automated centrifugal microfluidic system that depends on isothermal amplification technology. This system simplifies the nucleic acid extraction process and uses RT-LAMP to detect SARS-CoV-2. The experimental results are displayed by the change in the color of the fluorescent dye, and the entire experiment can be completed in less than 70 min (25). The above methods achieved sample-to-answer and met high-throughput detection. However, it is difficult to realize on-site and to self-test at home due to complex manufacturing processes and to those methods requiring precision instruments and professional personnel.

In this study, we constructed two independent modules of ALP FINA and SV RPA that are based on paper-based extraction and RPA-LFD, respectively, and integrated them into a system that could realize nucleic acid extraction, amplification, and visual detection. As to the ALP FINA module, we improved the paper-based nucleic acid extraction and replaced the enzyme lysis with alkaline lysis, which omits the steps of centrifugation and heating and is not dependent on equipment. Subsequently, the portable nucleic acid extraction chamber was designed to further simplify the operation procedure. The ALP FINA module not only reduces cost and simplifies the steps but also has an approximate extraction efficiency that is comparable to that of a traditional centrifugal column, endowing it with great advantages in field applications. The RPA-LFD technology can be operated at room temperature with impressive advantages, such as high sensitivity, good specificity, rapid reaction, and electricity-independence, making it ideal for field detection (26, 27). However, the main drawback of its traditional open-mode operation was the production of nucleic acid contamination. In order to solve this problem, we designed the SV RPA module, which integrated the amplification and detection into a sealed chamber, to avoid the aerosol contamination of nucleic acid. The commercially available AmpliVue system (QUIDEL, USA) combined the helicase-dependent isothermal amplification technique with the LFD to achieve

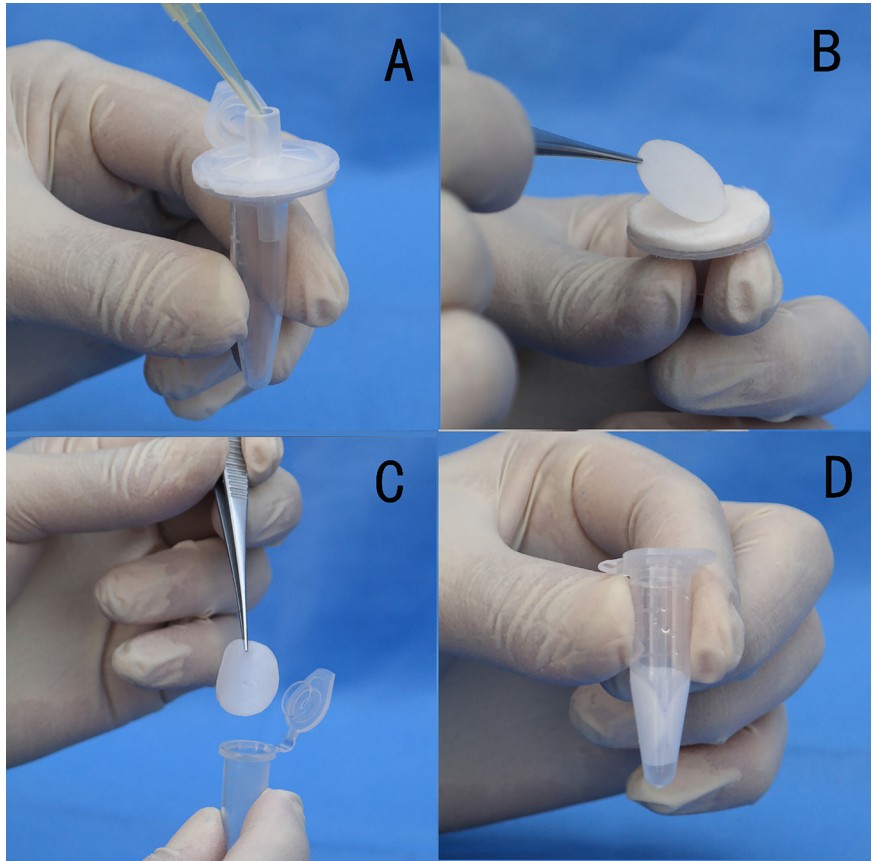

**FIG 5** The assembled ALP FINA prototype and ALP FINA operation. (A) Pipet the lysis reagent and 100% ethanol sequentially into the DNA capture membrane through the inlet port of the ALP FINA, and collect the waste solution from the waste outlet channel with a 1.5 mL tube. (B) Open the ALP FINA and separate the fusion 5 membrane from the cellulose filter paper. (C) Place the membrane in a clean 1.5 mL tube and leave to dry for 5 min. (D) Add 300 $\mu$L of nuclease-free water to elute the DNA.

integrated detection, but its application was limited due to poor sealing (28). Our integrated ALP FINA and SV RPA system showed good performance, could achieve nucleic acid extraction, amplification, and detection within 25 to 35 min, required no electricity or other laboratory infrastructure, and avoided the aerosol contamination of nucleic acids. In addition, the stability of the system at different environmental temperatures is crucial for its future applications under various geographical conditions. In this study, we analyzed the detection ability of the system at temperatures ranging from 25 to 45°C and demonstrated its reliability, thereby laying a foundation for its future applications.

Moreover, this type of modularized system provides additional flexibility, given that both modules are compatible with a variety of technologies (8, 29). For example, the SV RPA device can adopt not only the RPA but also the nucleic acid sequence-based amplification (NASBA), strand displacement amplification (SDA), and helicase-dependent amplification (HDA). Besides the alkali lysis method, the ALP FINA nucleic acid extraction device can also use the enzymatic extraction method for nasopharyngeal swab, blood, urine, feces, and other specimen types. In addition, each module can be used as a standalone device in conjunction with other technologies. For instance, the ALP FINA module could be used to prepare a sample for PCR and loop-mediated isothermal amplification (LAMP), and the SV RPA device could be used in conjunction with other nucleic extraction methods, such as centrifugal column and magnetic methods.

Among the 7 recognized HAdV species (A to G), species B and E are likely to cause acute respiratory infections and easily break out in special groups, such as kindergartens (27, 30), schools, and the army (22, 31). Therefore, the establishment of a rapid, accurate, and simple technique that is suitable for community and bedside diagnosis is

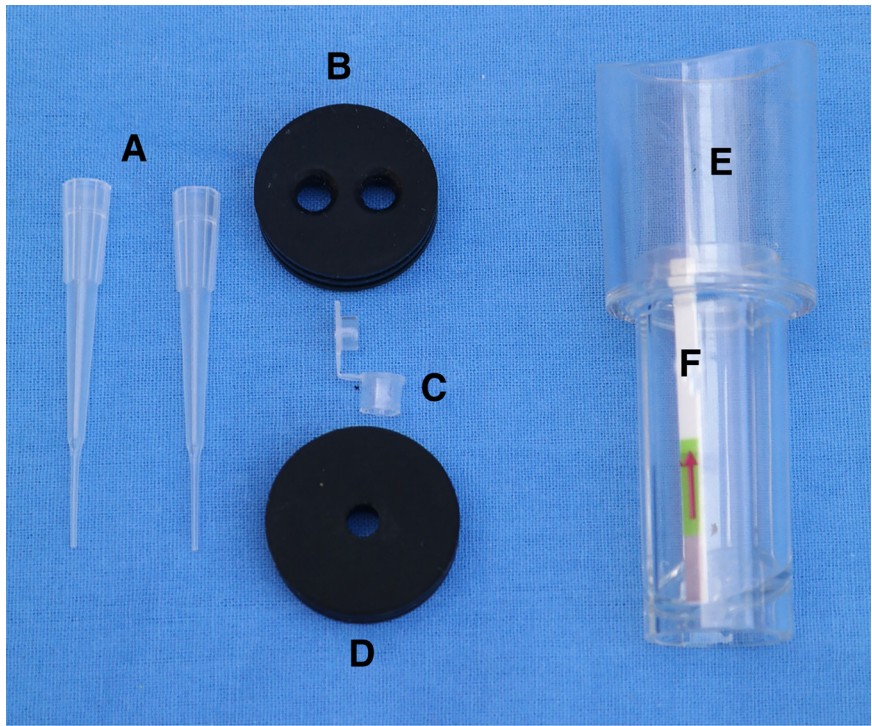

**FIG 6** Exploded view of the SV RPA procedure. (A) Reaction tube. (B) Loading rubber stopper. (C) Open/close switch. (D) Seal rubber stopper. (E) Storage column. (F) Lateral flow dipstick.

crucial for the prevention and control of epidemic situations. We developed the integrated system of ALP FINA-SV RPA for the detection of HAdV B and E, and it has the advantages of high sensitivity, good specificity, simple operation, fast response, visual detection, no contamination, and independence on electric power. Also, we demonstrated its suitability for on-site and field applications.

## MATERIALS AND METHODS

**Clinical specimens and standard panel dilutions.** Cultured 293 cells infected with AdvB, 14 AdvB-infected nasal swab specimens, and 5 AdvE-infected nasal swab specimens were originally obtained from the Fifth Medical Center of the Chinese PLA General Hospital. AdvB and AdvE plasmid standards, targeting the hexon gene from nucleotide 27 to 254, were produced by Sangon Biotech (Beijing, China).

A preliminary experiment demonstrated that the quantity of AdvB DNA that we obtained from 200 $\mu$L of the AdvB-infected cell suspension was about $2 \times 10^{12}$ copies/$\mu$L. The standard gradient dilutions, which contained AdvB nucleic acid ranging from $10^2$ to $10^9$ copies/$\mu$L, were performed via spiking into different amounts of nasopharyngeal swab specimens from healthy volunteers.

**Prototype device setup. (i) ALP FINA prototype**. The housing of the ALP FINA device is comprised of a top cover with a sample inlet channel that is 0.5 cm in diameter and a bottom cover with a waste outlet channel that is 1.0 cm in height (Fig. 4 and 5). The covers are constructed from PC2805 material. A nucleic acid extraction membrane is firmly sandwiched between the snapped top and bottom covers. The nucleic acid extraction membrane is composed of two layers, with a circular Fusion 5 filter paper (diameter, 1 cm; Whatman Inc., Florham Park, NJ, USA) sitting on top of a circular cellulose filter paper (diameter, 2 cm; Whatman GB003 blotting paper, catalog number 10426972). The two filter papers are tightly joined under an external force.

**(ii) SV RPA prototype.** The SV RPA device consists of a storage column, loading rubber stopper, seal rubber stopper, open/close switch, reaction tube, and LFD (Fig. 6 and 7). The storage column is made of PC2805 material and is composed of two cylindrical layers. The upper layer is a pressurized chamber (diameter, 3 cm; height, 4 cm) that is comprised of a seal rubber stopper, open/close switch, and the upper side of the loading rubber stopper. The bottom layer is a detection chamber (diameter, 2 cm; height, 5 cm) that is formed by the underside of the loading rubber stopper, flow channel, reaction tube, test strip slot, and LFD. The loading rubber stopper and seal rubber stopper are made of silica gel with a diameter of 3 cm and a thickness of 1 cm. The stopper consists of three-layered seal rings on the edge and holes that are 0.5 cm in diameter (one for the seal rubber stopper and two for the loading rubber stopper) in the center of the stoppers. The central hole of the seal rubber stopper is equipped with an open/close switch that is fabricated from a 200 $\mu$L EP tube, the bottom of which is cut with scissors. Depending on the experimental need, the lid of the switch could be kept open or closed. The two slots of the loading rubber stopper allow for the insertion of reaction

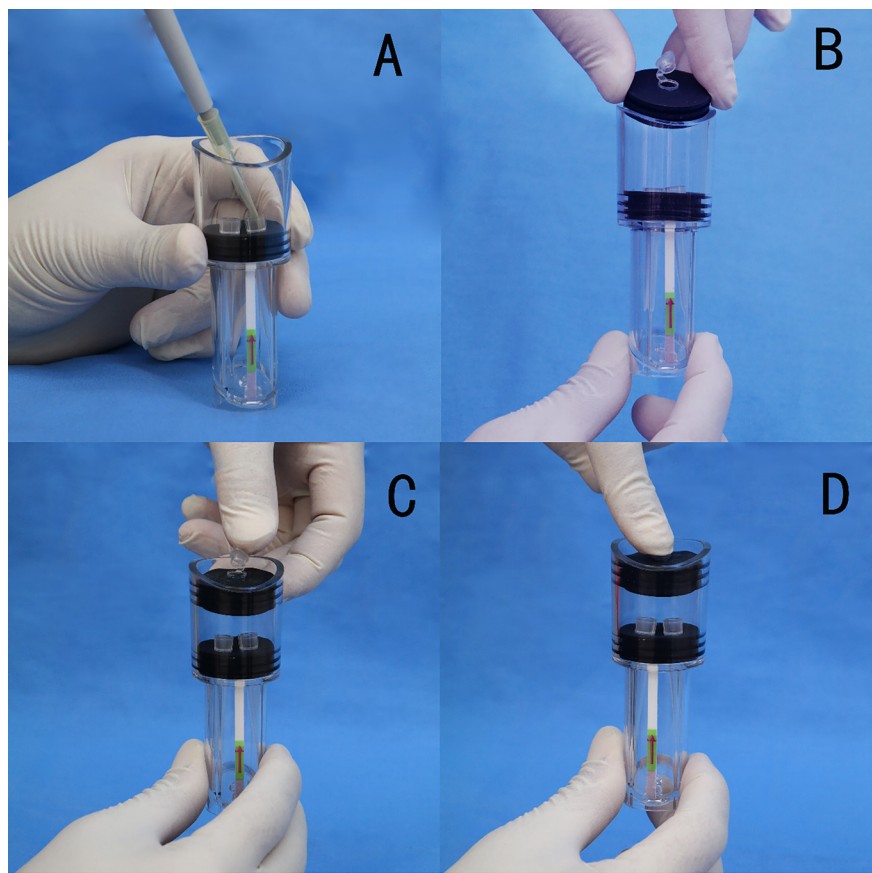

**FIG 7** The assembled SV RPA prototype and SV RPA operation. (A) Add RPA and LFD running buffer to reaction tubes. (B and C) Seal the device by closing the stopper and switch. (D) Perform the incubation at 37°C for 15 min. Apply pressure on the stopper to allow the liquid in the reaction tubes to flow out and come into contact with the LFD.

tubes with a structure resembling a pipette tip and a microsized pore at the bottom. The flow channel is placed at the bottom of the device with a 30° angle for the flow of the reaction mixture and contact with the LFD, and the LFD is fixed with the 3 mm× 5 mm test strip slot.

**Prototype device procedure. (i) ALP FINA operation.** The operation of the ALP FINA device for DNA extraction is described in Fig. 5. The procedure of DNA extraction is based on the alkaline lysis method, and all operations are performed at room temperature. First, 200 $\mu$L of the specimen were mixed with 500 $\mu$L of 10 mM NaOH to break down the cell wall, denature the protein, and release the genomic DNA. Then, the mixture was pipetted into the nucleic acid capture membrane through the inlet port of the extraction setup, and this was followed by the addition of 1 mL of 100% ethanol. The prevailing capillary forces generated by the blotter pad result in flow of the lysis reagent, which captures nucleic acid in the membrane and leaves the protein and other nonnucleic acid fragments outside. 100% ethanol was used to remove the nonnucleic acid fragments that bound to the membrane. Subsequently, the fusion 5 membrane was separated from the blotter, placed in a clean 1.5 mL tube, and left to dry for 5 min. Finally, 300 $\mu$L of nuclease-free water was added to the tube and incubated for 2 min to elute the DNA. The supernatant was transferred into another new tube in preparation for nucleic acid amplification.

**(ii) SV RPA operation**. Recombinase polymerase amplification and visual detection used the SV RPA prototype, as shown in Fig. 7. A typical 50 $\mu$L RPA (TwistAmp Exo Kit) was prepared in one of the reaction tubes, while the other was filled with 50 $\mu$L of the provided LFD running buffer. Each RPA reaction consisted of 29.5 $\mu$L of rehydration buffer, 0.6 $\mu$L of TwistAmp LF probe (10 $\mu$M), 2.1 $\mu$L of unlabeled forward primer (10 $\mu$M), 2.1 $\mu$L of biotin-labeled reverse primer (10 $\mu$M), 8.2 $\mu$L of nuclease-free water, 2.5 $\mu$L of magnesium acetate, and 5 $\mu$L of the extracted DNA, as described above. The liquid in the two reaction tubes was maintained by surface tension at the liquid-air interface, without flowing freely.

After the addition of the reaction components, we sealed the device by closing the switch and incubated the device at 35°C for 15 min. After the incubation, pressure was applied on the stopper to rapidly increase the air pressure in the pressurized chamber. As a result of the pressure increase, the RPA amplification solution and the LFD running buffer flowed out of the reaction tubes through the microsized pores into the detection chamber. These liquids could be collected at the slanted surface of the flow channel, mixed at the card slot, and reacted once in contact with the LFD.

**(iii) Complete prototype operation.** The ALP FINA and SV RPA modules were combined to

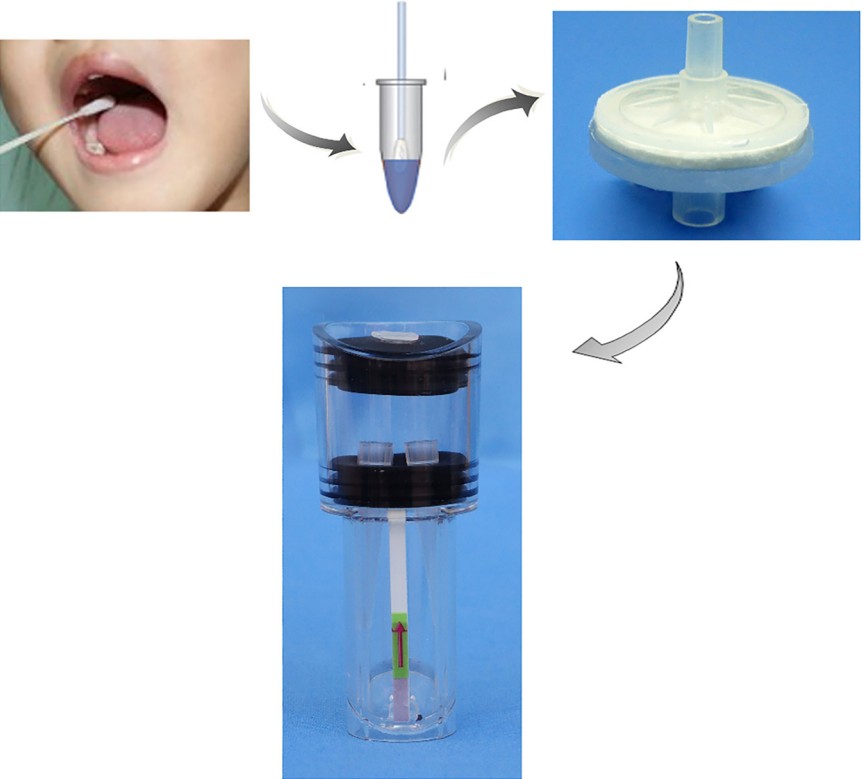

**FIG 8** The ALP FINA and SV RPA modules combined to complete DNA extraction, amplification, and detection steps.

complete the DNA extraction, amplification, and detection steps (Fig. 8). Nucleic acid extraction was carried out by adding the mixture of samples and NaOH, along with absolute ethanol, into ALP FINA. Subsequently, 5 $\mu$L of the extracted nucleic acid and other reaction component were added to the SV RPA module. After amplification, pressure was gently applied to the seal rubber stopper so as to allow the liquid in reaction tubes to flow out and come into contact with the LFD. Finally, a visible band on the LFD could be monitored by the naked eye, without the need of a sophisticated instrument (Fig. 2).

**Evaluation of the assay. (i) ALP FINA.** ALP FINA was compared to the traditional centrifugation extraction method using a QIAamp DNA Minikit (Qiagen GmbH, Hilden, Germany). Both methods were applied to extract DNA from equal volumes (200 $\mu$L) of reference strains with serial dilutions. Subsequently, the DNA extractions were quantified via quantitative PCR (qPCR), using ABI 7500 software v2.0 (Applied Biosystems, Foster City, CA, USA). In each qPCR run, the standard curve was generated via the calculation of the average cycle threshold value and a logarithm of the AdvB DNA standard dilution. The effective viral DNA concentration was quantified via standard curve differences.

**(ii) SV RPA.** The sensitivity of the SV RPA was evaluated using dilutions of AdvB and AdvE DNA plasmid standards and was compared with the results of qPCR. The specificity was confirmed with DNA samples from five other species (AdvC, AdvD, encephalitis virus, H1N1 virus, and norovirus). Given the importance of the reaction temperature and the time in point-of care testing (POCT), the optimal temperature was determined by performing the reaction in thermostats in temperatures ranging from 20°C to 50°C for 20 min each. The optimal reaction time was determined by stopping the reactions performed at 35°C at 0, 5, 10, 15, 20, and 25 min.

**(iii) Sealing verification.** To evaluate the sealing integrity of the SV RPA device, repeated experiments were performed in a separate room and compared to the traditional open-operation mode, referring to the standard operating procedures of molecular laboratories. Using $10^6$ copies/$\mu$L of AdvB DNA and nuclease-free water as the positive and negative controls, respectively, the experiment was repeatedly performed twice a day for five days under the two conditions described so as to monitor the occurrence of false-positive results.

**Primer and probe design.** The hexon gene of AdvB and AdvE from GenBank (GenBank no. AB330084, AB330085, AB330088, AB330092, AB330095, AB330097, AB330102, AB330115, AB330116, AB330131, and KF911353) were used for DNA detection as a target sequence. The universal primers for the simultaneous detection of both Adv types were designed using Primer 5.0, based on the alignment of these sequences and on the identification of type-specific conserved regions. Several primer pairs were evaluated, and one primer combination was eventually determined to be optimal. The primers for SV RPA and qPCR are described in Table 3. All of the oligonucleotides were produced by Sangon Biotech (Beijing, China).

**TABLE 3** List of primers and the probe for the SV RPA and qPCR, based on the AdvB and AdvE DNA

| Method | Name | Sequence (5′–3′) |
|---|---|---|
| RPA | RF | CATGCACATCGCCGGACAGGATGCTTCGGA |
| | RR | Biotin-TTTGTAAGAGTATGTATTGTCCTCCCGGTC |
| | Probe | FAM-TAGAAACCCCACAGTAGCGCCCACCCACGAT/THF/TGACCACCGACCGTAGC/C3-Spacer/ |
| PCR | RF | CATGCACATCGCCGGACAGGATGCTTCGGA |
| | RR | TTTGTAAGAGTATGTATTGTCCTCCCGGTC |

**Statistical analysis.** The statistical analysis was performed using the SPSS software package, version 21.0 (IBM). Probit regression was used to determine the limit of detection and its 95% confidence interval.

## ACKNOWLEDGMENTS

This work is partly funded by the Institute's own research projects and is partly self-financed. We wish to acknowledge the Fifth Medical Center of the Chinese PLA General Hospital for the cultured 293 cells infected with AdvB as well as the 14 AdvB-infected and 2 AdvE-infected nasal swab specimens.

K.S. and X.Y. designed and constructed the SV RPA sample testing system as well as the detection device. Y.W. designed and fabricated the ALP FINA nucleic acid extraction system. Q.G. collected the clinical samples. W.F. and C.Z. optimized the whole experiment. Q.L. and W.A. validated the clinical sample. Y.Z. repeatedly verified the stability of the system at temperatures of 40℃ as well as revised the manuscript. W.X. and D.X. supervised the project. K.S. and Y.W. wrote the manuscript with input from all other authors.

We declare that the research was conducted in the absence of any commercial or financial relationships that could be construed as a potential conflict of interest.

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
