## [Reviewer comments · Microbiology Spectrum]

Microbiology Spectrum

A novel sample-to-answer visual nucleic acid detection system for adenovirus detection

KUI SUN, Xiaodong Yang, Yanan Wang, Qun Guan, WENLIANG FU, Chao Zhang, Qing Liu, Wenzheng AN, Yongqi Zhao, WEIWEI XING, and Donggang Xu

Corresponding Author(s): WEIWEI XING, Beijing Institute of Basic Medical Sciences

Review Timeline:

Submission Date:	December 19, 2022
Editorial Decision:	January 10, 2023
Revision Received:	March 10, 2023
Accepted:	March 10, 2023

Editor: Donna Neumann

Reviewer(s): Disclosure of reviewer identity is with reference to reviewer comments included in decision letter(s). The following individuals involved in review of your submission have agreed to reveal their identity: Michael Owusu (Reviewer #1)

Transaction Report:

DOI: <https://doi.org/10.1128/spectrum.05170-22>

January 10, 2023

Dr. WEIWEI XING
Beijing Institute of Basic Medical Sciences
Taiping Road 27
Beijing
China

Re: Spectrum05170-22 (A novel sample-to-answer visual nucleic acid detection system for adenovirus detection)

Dear Dr. WEIWEI XING:

Link Not Available

Sincerely,

Donna Neumann

Journals Department
Reviewer comments:

Reviewer #1 (Comments for the Author):

Authors describe a Point-of-Care device that could be used for detection of Adenovirus B and E in resource limited settings. The technology appear good and workable in resource poor countries.

Authors should take note of minor comments:

1. Introduction: "control" should be "controlling the spread"
2. Authors should try and re-test their serially diluted samples in replicates and determine the limit of detection based on a 95%CI. They can use Probit regression to do this. This will give an idea about positive replicate samples that can be detected by the assay.
3. Authors should also give line number to the manuscript to help with review.

4. Although this is expected to work at room temperature, authors should note that some resource poor countries have high temperatures ranging up to 40 degrees celsius. Author may have to make recommendations for further field evaluations in real-world settings at selected resource poor countries. This will help test the robustness of this assay in these environments.

Staff Comments:

Preparing Revision Guidelines

Please return the manuscript within 60 days; if you cannot complete the modification within this time period, please contact me. If you do not wish to modify the manuscript and prefer to submit it to another journal, please notify me of your decision immediately so that the manuscript may be formally withdrawn from consideration by Microbiology Spectrum.

Editorial Office

Journal name Microbiology Spectrum

Dear editor:

Thank you for considering our manuscript ‘A novel sample-to-answer visual nucleic acid detection system for adenovirus detection’ by Sun et al (05170-22). We were pleased to receive your email with reviewer’ comments/suggestions and inviting us to submit a revised version of our manuscript. We would also like to thank the reviewers for their careful and insightful review of work that help us to further improve the manuscript. In this revised manuscript, we have carefully addressed all the comments/suggestions made by the reviewer by providing newly added experimental data and extensive editing. The amendments/adjustments to the manuscript for each of reviewers’ comments are marked in red. A detailed point-by-point response to all Reviewer’ comments can be found below.

We thank again for your kind consideration and look forward to hearing from you soon.

Sincerely

Donggang Xu

Beijing Institute of Basic Medical Sciences

Beijing, 100850, China.

Reviewer #1 (Comments for the Author):

Q1: Introduction: "control" should be "controlling the spread"

Response: We apologize for the typo issue in the original manuscript. We have revised this typo in the current manuscript and highlighted our revision in red.

Q2: Authors should try and re-test their serially diluted samples in replicates and determine the limit of detection based on a 95%CI. They can use Probit regression to do this. This will give an idea about positive replicate samples that can be detected by the assay.

Response: We are grateful for the insightful suggestion. According to CLSI EP17-A Protocols for Determination of Limits of Detection and Limits of Quantitation, we diluted the concentrations of plasmids of AdvB and AdvE to 0 copies/ μ L, 4 copies/ μ L, 6 copies/ μ L, 8 copies/ μ L, 10 copies/ μ L, 12 copies/ μ L, 14 copies/ μ L respectively, and repeated 24 times for each concentration. Then the Probit regression model was adopted to determine the limit of detection based on a 95% CI. The results were added in the Result section (Sensitivity and specificity) of the updated manuscript and were highlighted in red.

Q3: Authors should also give line number to the manuscript to help with review.

Response: We agree with the reviewer and added line number in the

updated manuscript.

Q4: Although this is expected to work at room temperature, authors should note that some resource poor countries have high temperatures ranging up to 40 degrees celsius. Author may have to make recommendations for further field evaluations in real-world settings at selected resource poor countries. This will help test the robustness of this assay in these environments.

Response: We deeply appreciate the thoughtful comment from the reviewer, and highly agree with the reviewer that it is essential to confirm the detection stability of our established system at more extreme ambient temperatures. Accordingly, we used a thermostat to simulate higher temperature conditions to evaluate the system's detection capabilities at 35°C, 40°C, 45°C, separately. The result showed that the detection system remained stable at temperatures ranging from 30-45°C (Fig 8b). Furthermore, we revalidated our device by detecting plasmid templates of AdvB and AdvE at 40°C, at which our system was still effective (as shown in Fig 1 below). Besides, we supplemented a description of the applicability of the device at different ambient temperatures in the Discussion section of the manuscript.

Fig 1 Repeated verification of the system at 40 degrees Celsius

March 10, 2023

Dr. WEIWEI XING
Beijing Institute of Basic Medical Sciences
Taiping Road 27
Beijing
China

Re: Spectrum05170-22R1 (A novel sample-to-answer visual nucleic acid detection system for adenovirus detection)

Dear Dr. WEIWEI XING:

Your manuscript has been accepted, and I am forwarding it to the ASM Journals Department for publication. You will be notified when your proofs are ready to be viewed.

Sincerely,

Donna Neumann
Editor, Microbiology Spectrum
